# The Utility of Contrast-Enhanced Mammography in the Evaluation of Bloody Nipple Discharge—A Multicenter Study in the Asian Population

**DOI:** 10.3390/diagnostics14202297

**Published:** 2024-10-16

**Authors:** Ann-Hui Jamie Ong, Yonggeng Goh, Swee Tian Quek, Premilla Gopinathan Pillay, Herng-Sheng Lee, Chen-Pin Chou

**Affiliations:** 1Department of Diagnostic Imaging, National University Hospital, Singapore 119074, Singapore; jamie.ong@mohh.com.sg (A.-H.J.O.); yong_geng_goh@nuhs.edu.sg (Y.G.); swee_tian_quek@nuhs.edu.sg (S.T.Q.); premilla_gopinathan_pillay@nuhs.edu.sg (P.G.P.); 2Department of Pathology and Laboratory Medicine, Kaoshiung Veterans General Hospital, Kaoshiung 813, Taiwan; hlee@vghks.gov.tw; 3Department of Medical Laboratory Sciences and Biotechnology, Fooyin University, Kaoshiung 813, Taiwan; 4Department of Radiology, Kaoshiung Veterans General Hospital, Kaohsiung 813, Taiwan

**Keywords:** contrast-enhanced digital mammography, pathological nipple discharge, bloody nipple discharge

## Abstract

Objective: To assess the efficacy of contrast-enhanced mammography (CEM) in differentiating benign from malignant breast lesions in Asian patients with bloody nipple discharge (BND). Methods: This retrospective study included 58 women with BND (mean age: 51.7 years) who underwent standardized CEM at institutions in Taiwan and Singapore. Lesion characteristics (size, enhancement, conspicuity, shape, margins) were evaluated on CEM by blinded radiologists. Non-enhanced mammography (MMG) and ultrasound (US) within a defined timeframe were compared for diagnostic accuracy. Benign or malignant status was confirmed by biopsy or 2-year imaging follow-up. Results: Malignancy was found in 29 of 58 lesions (50.0%), with ductal carcinoma in situ (DCIS) being the most common. CEM demonstrated a 100% negative predictive value (NPV) for non-enhancing lesions. Significant predictors of malignancy on multivariate analysis include enhancing lesions of size ≥ 1.5 cm (*p*-value 0.025) and suspicious morphological features (irregular/spiculated margins, irregular shape, segmental/linear NME distribution) (*p*-value < 0.001). CEM outperformed MMG (sensitivity: 58.6%) and US (sensitivity: 79.3%), achieving a sensitivity of 100% and the highest diagnostic accuracy at 81.3%. Additionally, a CEM size cut-off of 1.5 cm yielded a sensitivity of 73.5% and a specificity of 84.3%. Conclusions: CEM effectively differentiates benign from malignant lesions in patients with BND, improving diagnostic accuracy and potentially reducing unnecessary interventions.

## 1. Introduction

Nipple discharge is the third most common breast-related complaint, after breast pain and palpable lumps [1].

Physiological nipple discharge is characterized by bilaterality and typically occurs during pregnancy and lactation. Conversely, pathological nipple discharge is usually unilateral, uniductal, occurs spontaneously and can be either bloody or serous [1]. Most cases of pathologic nipple discharge are caused by benign conditions such as duct ectasia and papilloma [2], with reported risks of malignancy ranging from 5 to 15% and with ductal carcinoma in situ (DCIS) being the most common cause [3].

Among Asian women presenting to healthcare providers with nipple discharge, bloody discharge is the most frequently reported, making up to 60% of all nipple discharge cases in a recent Hong Kong study [4]. Often, bloody discharge prompts earlier consultation with healthcare professionals due to its alarming nature. In a meta-analysis by Chen et al. [5], patients presenting with bloody nipple discharge (BND) had a higher risk of malignancy when compared with patients present with non-bloody discharge. It is hence imperative to differentiate benign and pathological causes of BND for early detection of breast cancers and to avoid unnecessary surgical interventions for patients with benign causes [6,7].

Conventional diagnostic imaging plays an essential role in evaluation of BND; however, it exhibits inherent limitations. Ultrasound has limited sensitivity (56%) and specificity (75%) in detecting intraductal lesions associated with BND may lead to failures in identifying small or early-stage malignancies [8]. FFDM often shows negative results in up to 50% of patients with bloody nipple discharge [9]. While contrast-enhanced MRI shows higher sensitivity as compared to FFDM and US, it is deemed not to be cost-effective and generates additional false positive findings [10]. Ductography is technically challenging and is a potentially uncomfortable procedure for patients with reported failed procedure rates of 15–23% [9]. Ductoscopy, a minimally invasive procedure which allows direct visualization, shows potential benefits [11] but may not be capable of reaching the peripheral small branches due to the scope’s small diameter [12]. There is hence an unmet need for an optimal imaging modality for the evaluation of BND.

Contrast-enhanced mammography (CEM), an emerging technique, has been shown to be a potential cost-effective substitute to MRI in many aspects of breast imaging such as breast cancer detection and the staging of disease [13,14,15,16,17]. There is preliminary evidence from non-Asian populations suggesting that CEM demonstrates a high negative predictive value (NPV) in evaluation of patients with BND, and a normal CEM study could potentially exclude malignancy and limit the use of unnecessary surgical interventions [18]. However, these results may not be directly translatable to other population settings (e.g., the Asian population) with denser breast parenchyma which could render accurate imaging evaluation more challenging [19,20] To date, there are no studies which have evaluated the role of CEM in the evaluation of BND in the Asian population to the authors’ best knowledge. Hence, the goal of the study is to evaluate the feasibility and utility of CEM in evaluation of BND in the Asian population—in terms of its sensitivity, specificity and diagnostic accuracy.

## 2. Methodology 

This retrospective study included two cohorts of adult women (≥18 years old) who underwent diagnostic or screening CEM at the Department of Radiology, Kaohsiung Veterans General Hospital (KSVGH) (Kaohsiung, Taiwan), from February 2012 to November 2022 and the Department of Radiology, National University Hospital (NUH) (Singapore, Singapore), from December 2018 to January 2020. All patients underwent diagnostic FFDM and ultrasound followed by CEM examination—either requested as part of clinical investigations or of a previously performed prospective study [14]. The study was approved by both local institutional review boards (IRB KSVGH23-CT6-28 and NHG DSRB 2016/00508). Informed consent for participation was obtained from all subjects involved in the study.

### 2.1. Patient Selection

The cohorts were gathered systemically during the study period and a total of 810 CEM studies were collected from these two institutions. Patients with no documented history of bloody nipple discharge in their medical records at the time of CEM exam and patients with a history of papillomas which predispose them to bloody nipple discharge were excluded from the study. Finally, 58 eligible women with bloody nipple discharge were included in this study.

### 2.2. CEM Image Acquisition Protocol

All CEM images were acquired using a mammography system (Selenia Dimensions, Hologic, Marlborough, MA, USA) with dual-energy exposure. The contrast medium Omnipaque 350 (GE Healthcare Inc., Chicago, IL, USA) was injected into patients via an automatic power injector (Vistron CT injection system, Medrad, Warrendale, PA, USA) at a volume of 1.5 mL/kg of body weight and rate of 3 mL/s through a peripheral intravenous cannula. 

The CEM images were obtained following the completion of the contrast medium injection. Mediolateral oblique (MLO) and craniocaudal (CC) views of the symptomatic breast would be obtained first followed by CC and MLO views of the contralateral breast. Two exposures (i.e., a high-energy beam at 45–49 kVp and a low-energy beam at 26–32 kVp) were obtained almost simultaneously for each view and a subtracted image between the two was generated to visualize contrast enhancement of both breasts. The image acquisition of all four views was completed within 7 min. Both institutions utilized the same vendor equipment and protocol.

### 2.3. CEM Image Interpretation and Analysis

Image analysis of FFDM, US and the recombined CEM images was carried out by two breast radiology consultants with at least 10 years of experience in the field of breast imaging; the final diagnosis of the CEM images was reached upon consensus. 

Image interpretation was performed according to the contrast-enhanced mammography (CEM) Breast Imaging-Reporting and Data System (BI-RADS) lexicon 2022. The assessment began by detecting enhancing lesions and classifying them as either mass or non-mass enhancements with assessment of their conspicuity. Lesions classified as BI-RADS 1, 2 and 3 were considered negative, whereas those categorized as BI-RADS 4 and 5 were treated as positive.

In enhancing mass lesions, further assessment of the lesions’ margins (circumscribed or not circumscribed), shape (oval, round or irregular) and internal enhancement characteristics (homogeneous or heterogeneous) was performed. In cases of non-mass enhancement, further assessment of distribution (focal, linear, segmental, regional, multiregional or diffuse) and the pattern of internal enhancement (homogeneous, heterogeneous, clustered and clumped) was conducted. Lesion distribution (focal, linear, segmental or regional) and the extent of enhancement (partial, complete or beyond) were also assessed.

### 2.4. Pathological Diagnosis

The correlation between contrast enhancement in CEM images and histological results was utilized to categorize outcomes as either benign or malignant based on biopsy or surgical excision reports. In cases with histological confirmation (i.e., biopsy or surgical excision reports), correlations between CEM imaging parameters and histological results were performed to categorize outcomes as benign or malignant. In cases without histological confirmation, the presumed benign nature of imaging findings was validated by clinical evaluations and imaging follow-ups lasting over 24 months to ensure true benign status.

### 2.5. Data and Statistical Analysis

Statistical analyses were carried out using SPSS (Statistical Package for the Social Sciences) version 29 (IBM Corporation, Armonk, NY, USA). Numerical variables were summarized using means, standard deviations and range; categorical data were expressed as the frequency (count) and relative frequency (percentage). Differences in numerical variables were assessed using the two-sample *t* test; the chi-squared (χ^2^) test or Fisher’s exact test was used for categorical variables [21]. Standard diagnostic indices, including sensitivity, specificity, positive predictive value (PPV), negative predictive value (NPV) and diagnostic accuracy, were calculated as described by Galen [22]. The Receiver Operating Characteristic (ROC) curve was used to determine the optimal cut-off for size. The Youden Index, a summary statistic of the ROC curve, was used to identify the cut-off point that maximizes the sum of sensitivity and specificity [23]. Predictors of malignancy were analysed using logistic regression. Variables with *p* < 0.05 in the univariate logistic regression were included in the multivariate logistic regression. Statistical significance was set at two-sided *p* < 0.05. 

## 3. Results

### 3.1. Patient Demographics and Final Histology

This study included 58 patients who presented clinically with BND. Their ages ranged from 29 to 71 years (mean age: 51.7 ± 10.8 years) (Table 1). The patient characteristics and histological findings are summarized in Table 1. In total, 54 out of 58 lesions (93.1%) had histological correlation with biopsy or surgical records. The remaining 4 out of 58 (6.9%) with no histological correlation had negative or benign imaging findings which did not warrant an invasive biopsy/surgical procedure. Nonetheless, these four cases were followed up with imaging and clinical examination for at least 2 years to ascertain true benign status. All four patients had no persistent BND at their latest follow-up.

The different pathologies encountered in our study are illustrated in Table 1. In total, 29 out of 58 (50.0%) lesions were determined to be malignant as proven on histopathological analysis, while 29 (50.0%) were deemed benign (Table 1). Ductal carcinoma in situ (DCIS) was the most common malignant pathology, found in 16/58 cases (27.6%) while Papilloma was the most frequently represented benign pathology, found in 12/58 cases (20.7%). The age distribution in benign cases (29–71 years, mean 49.9 ± 10.8 years) did not significantly differ from that of malignant cases (35–70 years, mean 52.8 ± 10.3 years).

Among the benign pathology cases (*n* = 29), two were identified as high-risk lesions of atypical ductal hyperplasia (ADH) that required additional management. Initially discovered through ultrasound-guided biopsies and subsequently removed surgically, the final pathology reports for these ADH cases showed no evidence of disease upgrade. Of the 29 benign cases, 17.2% of patients (5/29) underwent surgery, while 82.8% of patients (24/29) were managed conservatively with routine follow-up care. Of these 24 patients, 83.3% (20/24) showed resolution of their symptoms while the remaining 16.7% (4/24) had intermittent symptoms.

### 3.2. Contrast-Enhanced Mammography Findings

In this study, CEM examinations were categorized into non-enhancing and enhancing groups based on contrast uptake patterns (Table 2). Among these 58 lesions, 11 out of 58 cases (19.0%) were classified as non-enhancing, and all were subsequently proven to be benign. In contrast, 47 out of 58 cases (81.0%) were categorized as enhancing. Of the enhancing lesions, 29 out of 47 cases (61.7%) were diagnosed as malignant, while 18 out of 47 cases (38.3%) were classified as benign. 

### 3.3. Predictors of Malignancy

Non-enhancement on CEM indicates 100% NPV for breast malignancy in patients with BND (Figure 1). The presence of enhancement on CEM, either mass or non-mass enhancement, showed significant correlation with malignancy (*p*-value < 0.001). Other significant predictors for malignancy include a low BPE level (minimal or mild), high or moderate lesion conspicuity, irregular or spiculated margins, irregular shape of enhancing masses and segmental or linear distribution of non-mass enhancements. The size of the lesion (see below) was also a significant predictor. 

Detection of malignancy was significantly higher in cases with low (minimal or mild) BPE levels, as compared to their counterparts with higher (moderate or marked) BPE levels. In total, 23 out of 35 cases (65.7%) with low BPE were histopathologically diagnosed with malignancy while 6 out of 23 cases (26.1%) with high BPE had histology-proven malignancy (*p*-value = 0.003). Similarly, enhancing lesions with high or moderate lesion conspicuity were more likely to be malignant. In total, 23 out of 30 cases (76.7%) with high/moderate lesion conspicuity had histology-proven malignancy while 6 out of 17 cases (35.3%) with low lesion conspicuity were found to be malignant on histopathological analysis. 

Of 22 mass enhancing lesions identified on CEM, 16 (72.7%) had an irregular shape, and 16 (72.7%) cases had irregular or spiculated margins. In total, 12 out of 16 cases (75%) with irregular shape were found to be malignant (*p*-value 0.003). In total, 12 of the 16 cases (75%) with irregular/spiculated margins were found to be malignant (*p*-value 0.003). 

Analysis of NME distribution on CEM in 25 cases revealed two predominant patterns: segmental or linear (*n* = 14, 56%) and other distributions encompassing focal, diffuse or regional patterns (*n* = 11, 44%). Notably, all cases (100%) with segmental or linear NME distribution were subsequently diagnosed as malignant. Conversely, the remaining NME patterns (focal, diffuse and regional) were associated with a mixed diagnosis, including three malignant cases (27.3%) and eight benign cases (72.7%). A segmental or linear NME distribution was found to be significantly associated with malignancy (*p*-value < 0.001).

The internal enhancement characteristics and the extent of enhancement did not show statistically significant associations with malignancy. 

### 3.4. Optimal Size on CEM for Malignancy Detection

Receiver Operator Characteristic (ROC) analysis was performed to evaluate the sensitivity and specificity of malignancy detection as CEM size increased (Figure 2). The optimal cut-off size for an enhancing lesion (determined using the Youden Index) was 1.55 cm (sensitivity 72.4%, specificity 86.2%, Youden index 0.586) and 1.75 cm (sensitivity 69.0%, specificity 89.7%, Youden index 0.586). A smaller CEM size of approximately 1.5 cm showed an increased sensitivity of 73.4% with a corresponding specificity of 84.8% (*p*-value = <0.001, Youden index 0.582). In our study population, there were no lesions between the sizes of 1.51 cm and 15.4 cm. Hence, similar results were achieved using a cut-off at either 1.5 cm or 1.55 cm. Lesions larger than the cut-off (1.5 cm and 1.75 cm) were more likely to represent malignancy (*p*-value < 0.001).

### 3.5. Multivariate Analysis of Significant Predictors

In order to identify the strongest independent predictors, multivariate analysis was performed via logistic regression. The results of the initial univariate and final multivariate analysis are found in Table 3. CEM enhancement was excluded from logistic regression analysis due to its 100% sensitivity. Morphological predictors, including segmental/linear NME distribution, the irregular/spiculated margins of enhancing masses and the irregular shape of enhancing masses were grouped together so as to achieve the same sample size for comparison as other variables significant on univariate analysis—namely BPE level, size of lesion and lesion conspicuity. 

Malignant morphology (i.e., the presence of any segmental/linear NME distribution, irregular/spiculated margins of enhancing masses and the irregular shape of enhancing masses) was a significant predictor of malignancy after multivariate logistic regression (Adjusted O.R. 120.1, 95% C.I. 8.9–1618.7, *p*-value < 0.001) (Figure 3). An enhancing lesion of a size ≥ 1.5 cm also remained a significant predictor of malignancy after multivariate logistic regression. (Adjusted O.R. 22.5, 95% C.I. 1.5–343.8, *p*-value 0.025.)

### 3.6. Comparing CEM with Ultrasound (US) and Full-Field Digital Mammography (FFDM)

#### 3.6.1. BIRADS Scoring

All 58 patients received combined evaluation of mammogram and US as part of routine imaging for BND. In total, 13 were deemed as BIRADS 3 or lower while 45 were deemed as BIRADS 4–5 after a combined evaluation of mammograms and US. This is illustrated in Table 4. Correlating with final histology, routine imaging with mammograms and US showed a sensitivity of 89.7%, a specificity of 34.5%, a PPV of 57.8% and an NPV of 76.9%.

For the 13 cases which were deemed BIRADS 3 or lower, there were 10 benign (true negatives) and 3 malignant (false negative) lesions. Out of the 10 patients with benign findings (true negatives), 2 patients had completely negative mammographic and sonographic findings with no suspicious lesions targetable for biopsies. These 2 patients were followed up for at least 2 years to ascertain the true benign status of the lesions and resolution of their symptoms. CEM showed no enhancement in these cases to concur with their benign statuses. 

Interestingly, there were three malignant cases of lesions classified as BIRADS 3 or lower by sono-mammography images in our dataset. Two cases were secondary to mammographic and sonographic occult cases of DCIS. The last case was a benign mass appearing on US with well-circumscribed margins deemed as a BIRADS 3 lesion. However, CEM demonstrated surrounding non-mass enhancement around the mass and the lesion was subsequently biopsied to represent DCIS. 

#### 3.6.2. Diagnostic Performance

The diagnostic performance of digital mammography (MMG), ultrasound (US), combined mammogram and ultrasound and contrast-enhanced mammography (CEM) were calculated individually and compared using Receiver Operator Characteristic (ROC) analysis (Figure 4). Our findings showed that CEM had the best sensitivity (100%), negative predictive value (100%), positive predictive value (72.5%) and diagnostic accuracy (81.3%). CEM demonstrated excellent diagnostic performance with an AUC of 0.810. This is illustrated in Table 5. 

## 4. Discussion

### 4.1. Summary of Important Results

The study involved 58 patients and the sensitivity, specificity, PPV and NPV of CEM in evaluation of BND are 100%, 62.1%, 72.5% and 100%, respectively. The optimal cut-off value for lesion size on CEM for detecting breast cancer was determined to be 1.5 cm (sensitivity 73.5%, specificity 84.3%). Significant predictors of malignancy on multivariate analysis include an enhancing lesion of size ≥ 1.5 cm (*p*-value 0.025) and suspicious morphological features (irregular/spiculated margins, irregular shape and segmental/linear NME distribution) (*p*-value < 0.001). CEM BIRADS demonstrated excellent diagnostic performance with an AUC of 0.810.

### 4.2. Benefits of CEM and Clinical Applications

Our results demonstrated that CEM can be a very useful adjunct tool in assessment of BND. CEM demonstrated higher sensitivity, specificity and diagnostic accuracy as compared to traditional modalities (combined MMG and US) (Table 4). In our study, CEM as an adjunct was able to detect three false negative cases from combined MMG and US (two mammographic/sonographic occult cancers and one cancer upgrade from a BIRADS 3 lesion). 

CEM’s high negative predictive value (NPV) of 100% could aid in the risk stratification of patients presenting with BND. This could imply that these less suspicious patients can be reassured and managed conservatively with routine imaging/clinical follow-up, reducing the unnecessary further need for surgical interventions to achieve histological diagnosis. This will be helpful in cases of negative imaging findings or in cases with probably benign (BIRADS-3) findings. However, conventional diagnostic methods for BND are still necessary for BND patients with a high clinical suspicion of malignancy even if the findings on CEM are negative or benign.

In contrast, imaging features on CEM that raise suspicion for malignancy (e.g., high/moderate lesion conspicuity, irregular/spiculated margins of enhancing masses and a segmental/linear distribution of non-mass enhancing lesions) would prompt further interventions to achieve a histological diagnosis. These imaging features would also boost the confidence of diagnostic radiologists when assessing for imaging–pathology concordance after biopsy results to reduce potential false negatives. 

Ultimately, the authors would like to state that MRI remains the gold standard and excels in providing comprehensive 3D breast imaging and dynamic scanning sequences. However, it is often limited by its availability (long waiting times), long procedure time, and its high costs render this a non-cost-effective study to be performed for all patients with BND. On the other hand, CEM has a much shorter imaging acquisition time and can often be performed during the same patient visit to enable a one-stop work-up. Hence, CEM could represent a more logistical and cost-effective alternative for patients presenting with BND.

### 4.3. Lesion Conspicuity 

For lesions which enhance, CEM demonstrated a respectable PPV of 62.0%. For lesion conspicuity, we found that lesions which demonstrate higher conspicuity had a higher chance of breast malignancy as compared to lesions which demonstrate low conspicuity (*p*-value = 0.019). This is likely based on the phenomenon that neoplasms induce angiogenesis for further tumor growth [24]. It is important to note that the current assessment of conspicuity in our study (based on ACR guidelines) remain subjective and further studies with use of quantitative measures or software could further help to improve malignancy risk stratification in the future [25].

### 4.4. Morphpological Characteristics

Worrisome features on CEM, such as irregular margins (for enhancing masses) and worrisome distribution (linear/segmental for NMEs), show an increased chance of breast malignancies in patients presenting with BND. This is largely unsurprising as these are identical predictors of malignancy on breast MRI. These features are readily apparent in sizeable lesions on CEM but can be difficult to fully characterize in lesions which are smaller (<1 cm). Smaller lesions were also more likely to represent benign lesions in our study. We performed a Receiver Operator Characteristic (ROC) analysis and demonstrated that the optimal size cut-off in this study was 1.5 cm. Lesions ≥ 1.5 cm showed the high sensitivity and appropriate specificity for malignancy detection, with 21 out of 25 cases (84.0%) being identified as malignant.

### 4.5. Background Parenchymal Enhancement (BPE)

To fully evaluate lesions on CEM, it is ideal to evaluate the lesion in its entirety with no obscuration. These lesions are typically best evaluated in cases with minimal–mild BPE as moderate–marked BPE has been well-known to hinder accurate evaluation on contrast-enhanced modalities such as MRI [26]. Greater BPE has been reported to have a higher probability of developing breast cancer in high-risk women [27]. However, in our study, we did not find any relationships between moderate–marked BPE on CEM with breast malignancy. In contrast, detection of malignancy was significantly higher in cases with low (minimal or mild) BPE levels, as compared to their counterparts with higher (moderate or marked) BPE levels in our study. This is likely due to improved visibility of lesions on low BPE. The authors suggest that the relationship between BPE and malignancy may differ across specific clinical situations and patient groups. Further studies are warranted to understand the implications of BPE in patients presenting BND.

### 4.6. Limitations

Our study has a few limitations. Firstly, our small sample size restricts the statistical power of our analysis. Additionally, the homogeneity of our sample may limit the diversity of perspectives represented, potentially overlooking important nuances or variations within the population. Secondly, there is likely a selection bias in our cohort with a significant proportion of malignant lesions identified (50%). As CEM is a minimally invasive procedure with injection of iodinated contrast, women with mild symptoms would be less willing to undergo a minimally invasive procedure for a trivial condition. Therefore, there is a natural selection bias towards willing patients with more severe symptoms who are more likely to present with breast malignancies. However, the enriched cohort study was focused on the utility of CEM in assessment of BND and an equal distribution of malignant and benign conditions may have been useful in improving the understanding of CEM appearances for both benign and malignant causes of BND. The authors acknowledge that the cohort in this study may not be representative and we are currently actively increasing the number of patients at two different medical centers to strengthen our findings. Future studies will help us pinpoint and understand potential limitations and pitfalls. Future research endeavors may also include comparisons of MRI and CEM for BND when conventional diagnostic mammogram and ultrasound are negative. Despite these limitations, our study serves as a crucial foundation for further investigations into this area of inquiry.

## 5. Conclusions

The study shows CEM to be a valuable diagnostic tool for distinguishing between benign and malignant BND lesions, suggesting its potential utility in clinical practice. Future work should validate these results in larger studies and assess the integration of CEM with other diagnostic methods.

## Figures and Tables

**Figure 1 diagnostics-14-02297-f001:**
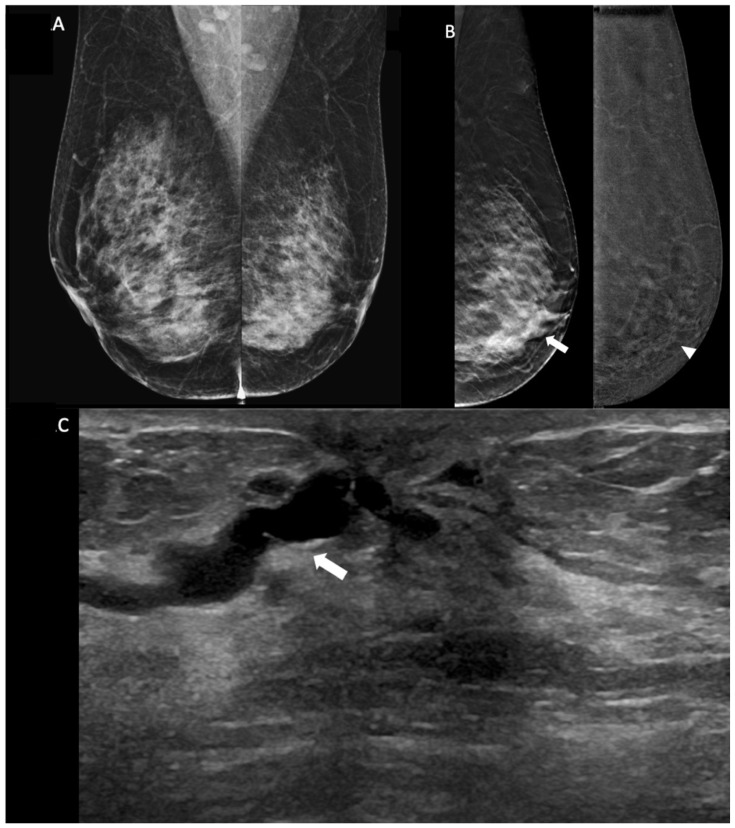
Images of a breast papilloma in a 65-year-old woman presenting with bloody nipple discharge. (**A**) Bilateral mammogram (MLO view): No suspicious mass or microcalcifications seen. Breast tissue is heterogeneously dense. A focal asymmetry is noted in the left breast, lower inner quadrant. (**B**) Digital breast tomosynthesis (**right**) and contrast-enhanced mammography (**left**) of the left breast (MLO view) demonstrate a focal asymmetry with increased density (arrow) in the lower inner quadrant. No abnormal contrast enhancement is seen in this corresponding area on CEM (arrowhead). (**C**) Ultrasound image of the left breast shows ductal dilatation (arrow) in the subareolar region.

**Figure 2 diagnostics-14-02297-f002:**
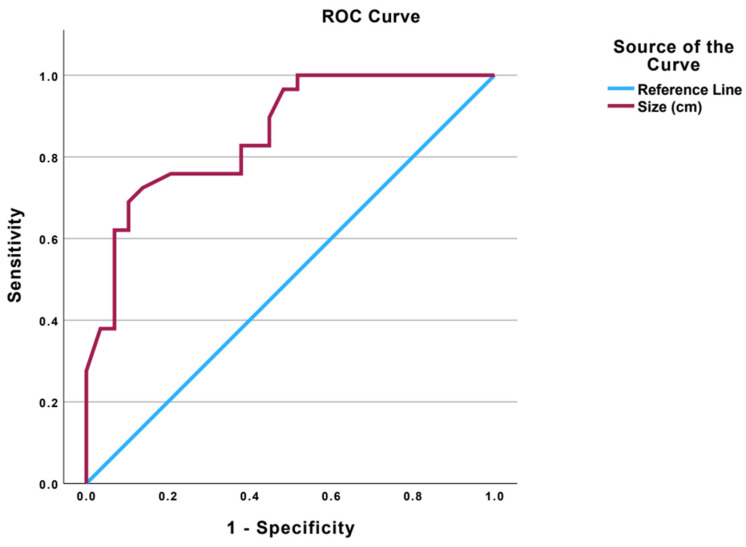
Analysis of an optimal size on CEM for malignancy detection.

**Figure 3 diagnostics-14-02297-f003:**
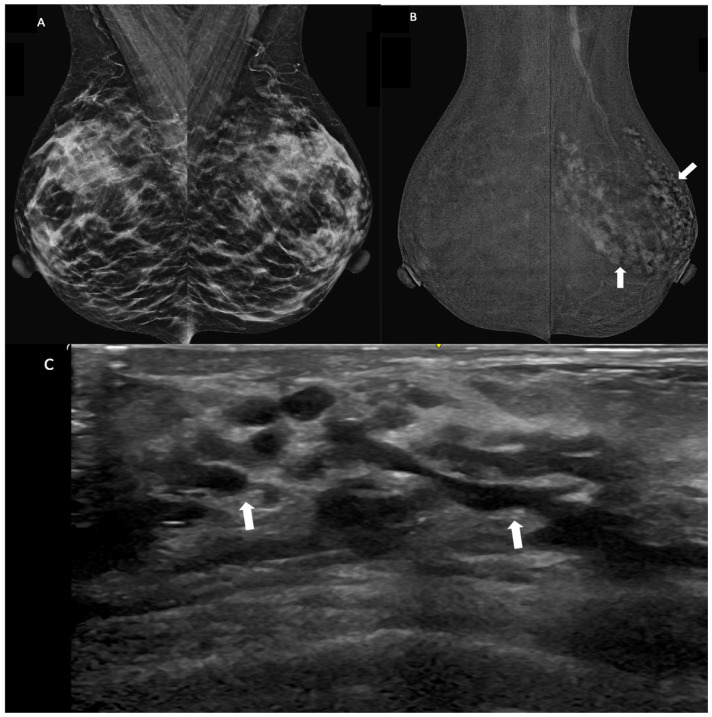
Images of a breast ductal carcinoma in situ (DCIS) in a 47-year-old woman presenting with bloody nipple discharge. (**A**) Bilateral mammogram (MLO view): No suspicious findings identified. Breast tissue is heterogeneously dense. (**B**) Contrast-enhanced mammography of the bilateral breasts (MLO view) reveals segmental non-mass enhancement (arrows) in the upper outer quadrant of the left breast. (**C**) An ultrasound of the left breast demonstrates ductal dilation (arrows) in the area corresponding to the segmental non-mass enhancement seen in contrast-enhanced mammography.

**Figure 4 diagnostics-14-02297-f004:**
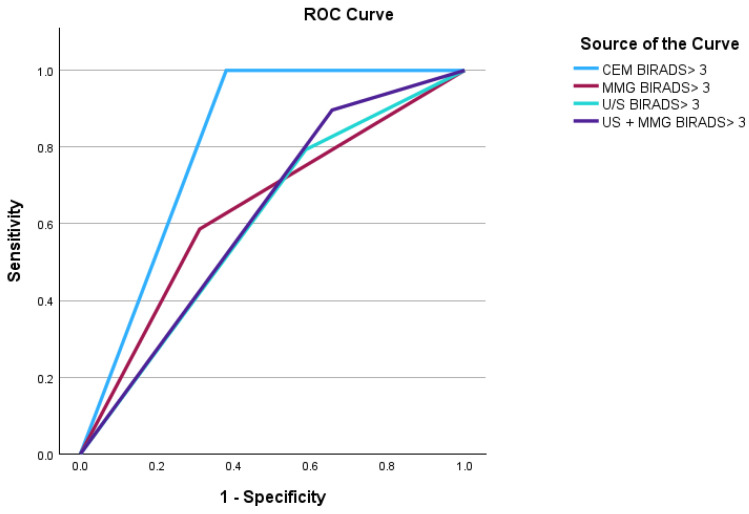
Receiver Operator Characteristic (ROC) curves of mammogram, ultrasound, mammogram + ultrasound and CEM.

**Table 1 diagnostics-14-02297-t001:** Patient characteristics and final histology.

Parameter	Value
**Age (years)**	
Mean ± SD	51.7 ± 10.8 years
Range	29 to 71
**Breast composition**	
Almost entirely fatty	0 (0.0%)
Scattered densities	10 (17.2%)
Heterogeneously dense	36 (62.1%)
Extremely dense	12 (20.7%)
**Malignancy (*n* = 29)**	
Ductal carcinoma in situ	16 (55.2%)
Invasive ductal carcinoma	11 (37.8%)
Others (e.g., invasive mucinous carcinoma, metaplastic carcinoma)	2 (6.9%)
**Benign disease (*n* = 29)**	
Papilloma	12 (41.4%)
Fibrocystic change	6 (20.7%)
Atypia (e.g., atypical ductal hyperplasia, flat epithelial atypia)	3 (10.3%)
Fibroadenoma	2 (6.9%)
Mastitis	2 (6.9%)
No biopsy (negative or benign imaging features, stable on follow-up)	4 (13.8%)

Patient characteristics and histological findings (*N* = 58).

**Table 2 diagnostics-14-02297-t002:** Correlation between CEM features and final histology.

CEM Features	Malignant		Benign		*p*-Value
	Count	%	Count	%	
**BPE level (*n* = 58)**					
Minimal	14	24.1	7	12.1	0.018
Mild	9	15.5	5	8.6	
Moderate	5	8.6	9	15.5	
Marked	1	1.7	8	13.8	
**Location of lesion (*n* = 58)**					
No lesion detected	0	0.0	3	5.2	0.353
Subareolar	19	32.8	18	31.0	
Peripheral	10	17.2	8	13.8	
**Presence of enhancement (*n* = 58)**					
Non-enhancing	0	0.0	11	19.0	<0.001
Enhancing	29	50.0	18	31.0	
**Extent of enhancement (*n* = 47)**					
Partial	6	12.8	5	10.6	0.726
Complete/beyond	23	48.9	13	27.7	
**Internal enhancement characteristics (*n* = 47)**					
Homogenous	6	12.8	2	4.3	0.692
Heterogenous or clumped	23	48.9	16	34.0	
**Lesion conspicuity (*n* = 47)**					
Low	6	12.8	11	23.4	<0.001
High or moderate	23	48.9	7	14.9	
**Size of enhancing lesion on CEM (*n* = 47)**					
<1.50 cm	8	17.0	14	29.8	<0.001
≥1.50 cm	21	44.7	4	8.5	
<1.75 cm	9	19.1	15	31.9	<0.001
≥1.75 cm	20	42.6	3	6.4	
**Shape of enhancing masses on CEM (*n* = 22)**					
Round/oval	0	0.0	6	27.3	0.003
Irregular	12	54.5	4	18.2	
**Margins of enhancing masses on CEM (*n* = 22)**					
Circumscribed	0	0.0	6	27.3	0.003
Irregular or Spiculated	12	54.5	4	18.2	
**Distribution of NMEs on CEM (*n* = 25)**					
Segmental or linear	14	56.0	0	0.0	<0.001
Others (e.g., focal, diffuse, regional)	3	12.0	8	32.0	

**Table 3 diagnostics-14-02297-t003:** Univariate and multivariate analysis of significant predictors. * For Marked BPE, OR = 1.0 is the reference category. ** A size cut-off of 1.5 cm was chosen for our study. Please see “Discussion” for further explanation.

Predictors	Unadjusted O.R. (95% C.I.)	*p*-Value	Adjusted O.R. (95% C.I.)	*p*-Value
** *BPE* **				
Minimal	16.0 (1.7–154.6)	0.017	19.0 (0.2–1588.8)	0.192
Mild	14.4 (1.4–150.8)	0.026	5.6 (0.1–395.6)	0.431
Moderate	4.4 (0.4–46.5)	0.213	18.8 (0.2–1441.3)	-
Marked *	1.0	0.043	1	0.517
** *High/moderate lesion conspicuity* **	10.6 (3.0–33.8)	<0.001	1.3 (0.1–18.0)	0.839
***Size of enhancing lesion ≥ 1.50 cm* ****	16.4 (4.3–62.2)	<0.001	22.5 (1.5–343.8)	0.025
** *Malignant morphology* **	84.4 (14.2–501.5)	<0.001	120.1 (8.9–1618.7)	<0.001

**Table 4 diagnostics-14-02297-t004:** Correlation between BIRADS and final pathology.

Imaging Modality and BIRADS	Malignant		Benign		*p*-Value
	Count	%	Count	%	
**Full-field digital mammography (FFDM)**					
BIRADS ≤ 3	12	20.7	20	34.5	0.035
BIRADS 4 or 5	17	29.3	9	15.5	
**Ultrasound (US)**					
BIRADS ≤ 3	6	10.3	12	20.7	0.089
BIRADS 4 or 5	23	39.7	17	29.3	
**Combined US + FFDM**					
BIRADS ≤ 3	3	5.2	10	17.2	0.028
BIRADS 4 or 5	26	44.8	19	32.8	
**Contrast-enhanced mammogram (CEM)**					
BIRADS ≤ 3	0	0.0	18	31.0	<0.001
BIRADS 4 or 5	29	50.0	11	19.0	

**Table 5 diagnostics-14-02297-t005:** Diagnostic Performance Metrics of Mammogram, Ultrasound, Mammogram + Ultrasound and CEM.

	Area Under Curve (AUC)	Sensitivity	Specificity	NPV	PPV	Diagnostic Accuracy
MMG	0.638	58.6	69.0	62.5	65.4	63.8
US	0.603	79.3	41.4	66.7	57.5	60.3
MMG + US	0.621	89.7	34.5	76.9	57.8	62.1
CEM	0.810	100	62.1	100	72.5	81.3

## Data Availability

The raw data supporting the conclusions of this article will be made available by the authors on request.

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
