# Peer review of "The Utility of Contrast-Enhanced Mammography in the Evaluation of Bloody Nipple Discharge—A Multicenter Study in the Asian Population"

_diagnostics, 2024, doi:10.3390/diagnostics14202297_

Round 1
Reviewer 1 Report
Comments and Suggestions for Authors
This study is well-conducted and well-written, presenting valuable data on the utilization of contrast-enhanced mammography (CEM) in evaluating bloody nipple discharge in the Asian population. There are no major concerns to address in this paper. However, the authors should consider the following minor revisions to improve clarity:
1. Please provide a brief explanation of the Youden index in the statistical analysis section of the methods part. This will help readers understand its application and significance in the context of the study.
2. In line 323, please edit 'CEDM' to 'CEM'.
Overall, this paper makes a significant contribution to the field and will be of interest to clinicians and researchers involved in breast imaging.
Author Response
REVIEWER 1
1) General feedback: This study is well-conducted and well-written, presenting valuable data on the utilization of contrast-enhanced mammography (CEM) in evaluating bloody nipple discharge in the Asian population. There are no major concerns to address in this paper. However, the authors should consider the following minor revisions to improve clarity… Overall, this paper makes a significant contribution to the field and will be of interest to clinicians and researchers involved in breast imaging.
Reply: The authors would like to thank the reviewer for the positive comments.
2) Please provide a brief explanation of the Youden index in the statistical analysis section of the methods part. This will help readers understand its application and significance in the context of the study.
Reply: The authors have inserted a new statement to explain Youden index under the statistical section. The sentence can be found in lines 158-160 and is read as the following:
“The Youden Index, a summary statistic of the ROC curve was used to identify the cut-off point that maximizes the sum of sensitivity and specificity [21].”
[21]: Ruopp MD, Perkins NJ, Whitcomb BW, Schisterman EF. Youden Index and optimal cut-point estimated from observations affected by a lower limit of detection. Biom J 2008; 50:419-430].
3) In line 323, please edit 'CEDM' to 'CEM'.
Reply: The authors apologize for the mistake. CEDM has been changed to CEM in line 358.
Reviewer 2 Report
Comments and Suggestions for Authors
Page 1, line 37. Consider using more recent references for the introductory statements.
Page 1, line 38-39. This sentence is subjective. You may consider, instead, describing pathologic (unilateral, single duct, spontaneous, bloody or clear) versus non-pathologic nipple discharge (bilateral, nonspontaneous) and then reporting the likelihood of malignancy for pathologic nipple discharge as well as the most common causes of nipple discharge.
Page 1, line 42. Please expand on what is meant by potential for misdiagnosis and difficulties in imaging evaluation.
Page 1, line 44. A patient with symptomatic BND will typically undergo biopsy or surgical consultation, for both benign pathology (such as papilloma) and malignancy, in efforts to find resolution for the patient’s BND. The introduction should create an argument for how recognition of benign pathology on CEM will aid in patient management. Even with benign pathology, patients with BND will undergo surgery to resolve symptoms.
Page 2, lines 47-56. This paragraph better describes the argument for evaluating CEM as a modality for evaluation of BND. I would suggest eliminating the prior surgical argument and expanding on the limitations of FFDM, US, MRI, and galactography.
Page 2, line 57. Review and edit style and word choice. For example, emerging and new are similar descriptors, so eliminate one of those words. Similar examples throughout (example page 2, line 60, population and groups).
Page 2, line 62. BND, alone, is a reason for surgical intervention even with benign pathology.
Page 2, line 67. Please specifically define the goal of this study.
Page 2, line 71. Patients with BND should not undergo screening mammography, as these patients are symptomatic. Please explain why screening CEM exams were included in the search for the study population.
Page 2, Patient selection. Duplicate/repeated phrases, such as “for both institutions/both centers”. I would suggest decreasing word count and including high yield information more succinctly.
Page 2, line 87. CEM was performed at two institutions, so briefly state if both institutions utilized the same vendor equipment and protocol.
Page 2, line 93. The protocol states CEM was performed first on the side of the “lesion of concern”. Was CEM performed first on the symptomatic side with BND or was there already a known “lesion of concern” prior to performing CEM?
Page 3, line 101. Was interpretation done by a radiologist or consultant? Please clarify credentials of the interpreter.
Page 3, CEM image interpretation and analysis paragraph. I would suggest reviewing these two paragraphs for style, word choice, grammar and punctuation. Similar recommendation for the “Pathology diagnosis” paragraph on page 3, line 117.
Page 3. Table 1. The majority of patients with BND have benign pathology, as you also mentioned in the introduction. Please explain why your population of BND patients is 50% malignancy. This would suggest a bias in the study population.
Page 4, line 145. Please clarify if the 4 cases who had follow-up only had persistent BND for the follow-up time period.
Page 5, Figure 1. (A) states no suspicious finding on mammogram but then describes a focal asymmetry. Please make certain to ensure accuracy of figure descriptions.
Page 7, line 180. For the 11 cases that were non-enhancing, please clarify what modality identified a lesion. Where any of these papillomas that were the cause of nipple discharge? Did they require excision to resolve the BND? As a reader, I would like to know how many of the benign cases of BND had resolution of symptoms during this study period.
Page 7, Table 2. Again, the large number of spiculated/irregular masses and segmental NME in this study population suggests possible selection bias, given that most cases of BND are typically benign. Did the study population have standard of care diagnostic FFDM and US prior to CEM? Is it possible that cases from the institutions were more likely to have malignancy based on imaging findings prior to CEM, and that is why CEM was performed? If so, please clarify in methods and patient selection. This would also explain the larger size (>1.5 cm) of lesions in this study. Lesions greater than 1.5 cm are typically seen on ultrasound, so it would help the reader if the methods of patient selection were clarified.
Page 7, line 189. Low BPE level being a predictor of malignancy is contradictory to high BPE increasing likelihood of malignancy. Could the low BPE level be a correlation because lesions are easier to detect on CEM with low BPE. This would be an item for discussion, and possible limitation of CEM.
Page 9. I would limit results in this manuscript for additional findings, such as CEM size correlation with final pathological size and un-enhanced mammographic morphologic predictors of disease. The results section should focus most attention on the primary and dominant secondary endpoints.
Page 10, line 268. When did the study patients have mammogram and ultrasound in relation to the timing of the CEM? Who interpreted the mammogram and ultrasound? The consultants or were charts reviewed? This portion of the study was not detailed in the Methodology section and should be described. Also, clarify whether screening or diagnostic mammography was performed (prior comment regarding screening versus diagnostic CEM as well).
Page 10, line 282 and 284. I would suggest using BI-RADS terminology throughout. For example, mass rather than nodule.
Page 11, line 303. There are a high number of malignant lesions in this study population, and this should be addressed in this study and any limitations to study design disclosed.
Page 11, line 305. One of the most important study results gleaned from this study population of symptomatic patients with BND is that all non-enhancing findings were benign. I would suggest expanding on this important detail. Surgical excision may still need to be performed due to symptoms, but it is helpful to know the high likelihood of benignity. Unfortunately, the selection bias of this study may hinder results as there are not a sufficient number of benign cases to make strong conclusions.
Page 11, line 326. Surgical management is appropriate for BND, even in benign cases to resolve the BND. Conservative management with imaging only follow-up may not be appropriate if BND is not resolved. The conclusions drawn here may be inappropriate.
Page 12, lines 340-355. I would consider deleting this paragraph which does not discuss the implications of results from this study.
The Discussion should be significantly shortened and reviewed for relevance and discuss only conclusions that can be drawn from this study as well as for style, grammar, and repetitive topics. The discussion should provide the primary and secondary findings of this study, comparison of this study to other relevant literature, and discussion of limitation and future research. This discussion section draws several conclusions unrelated to the study results.
Page 13, line 406. Future endeavors may include comparison of MRI and CEM for BND when conventional diagnostic mammogram and ultrasound are negative.
Page 13, line 411. The conclusion made regarding clinical utility of CEM for BND may not be appropriate because both benign and malignant BND may be managed surgically.
Comments on the Quality of English Language
This manuscript should be thoroughly reviewed for word choice, grammar, punctuation, unnecessary capitalization, and repetition.
Author Response
REVIEWER 2
1) Page 1, line 37. Consider using more recent references for the introductory statements.
Reply: The authors have added in a more recent reference for the introductory statement with a reference from Dengel et. al. in year 2022. It is now cited in the article with changes highlighted in the manuscript (Line 38).
2) Page 1, line 38-39. This sentence is subjective. You may consider, instead, describing pathologic (unilateral, single duct, spontaneous, bloody or clear) versus non-pathologic nipple discharge (bilateral, nonspontaneous) and then reporting the likelihood of malignancy for pathologic nipple discharge as well as the most common causes of nipple discharge
Reply: The authors have amended the introduction to include a more thorough discussion of pathologic nipple discharge to reduce subjectivity according to the reviewer’s comments. As pathologic nipple discharge includes discharges of various colours, the authors have also sought to report the likelihood of malignancy of non-bloody vs bloody nipple discharges. The following sentences have been amended in lines 37-60:
“Nipple discharge is the third most common breast-related complaint, after breast pain and palpable lumps [1]. It is especially alarming when the discharge is bloody, which may result in a high degree of anxiety in women because of fear of breast cancer. However, the majority of bloody nipple discharge (BND) is caused by benign conditions such as duct ectasia and papil-loma. The risk of underlying breast malignancy is low, with a 5-23% range related to the potential for misdiagnosis and difficulties in imaging evaluation.
Hence, BND presents as a clinical and diagnostic challenge for breast surgeons and radiologists. It is important to differentiate benign and pathological causes of BND for early detection of breast cancers and to avoid unnecessary surgical interventions for pa-tients with benign causes
Physiological nipple discharge is characterized by bilaterality and typically occurs during pregnancy and lactation. Conversely, pathological nipple discharge is usually unilateral, uniductal, occurs spontaneously and can be either bloody or serous [1]. The majority of pathologic nipple discharge is caused by benign conditions such as duct ectasia and papilloma [2], with reported risks of malignancy ranging from 5-15% and ductal carcinoma in-situ (DCIS) as the most common cause [3].
Among Asian women presenting to healthcare providers with nipple discharge, bloody discharge is the most frequently reported, making up to 60% of all nipple discharge cases in a recent Hong Kong study [4]. Often, bloody discharge prompts earlier consultation with healthcare professionals due to its alarming nature. In a meta-analysis by Chen et. al [5], patients presenting with bloody nipple discharge (BND) had a higher risk of malignancy when compared with patients present with non-bloody discharge. It is hence imperative to differentiate benign and pathological causes of BND for early detection of breast cancers and to avoid unnecessary surgical interventions for patients with benign causes [6,7].”
[1]: Vavolizza, R. D., & Dengel, L. T. (2022). Management of nipple discharge. Surgical Clinics of North America, 102(6), 1083-1094. https://doi.org/10.1016/j.suc.2022.06.006).
[2]: Vargas HI, Romero L, Chlebowski RT. Management of bloody nipple discharge. Curr Treat Options Oncol 2002; 3:157-161
[3]: Gupta D, Mendelson EB, Karst I. Nipple Discharge: Current Clinical and Imaging Evaluation. American Journal of Roentgenology 2020; 216:330-339).
[4]: Kan WM, Chen C, Kwong A. Implications of nipple discharge in Hong Kong Chinese women. Hong Kong Med J 2018; 24:18-24].
[5]: Chen L, Zhou WB, Zhao Y, et al. Bloody nipple discharge is a predictor of breast cancer risk: a meta-analysis. Breast Cancer Res Treat 2012; 132:9-14)
[6]: Montroni I, Santini D, Zucchini G, et al. Nipple discharge: is its significance as a risk factor for breast cancer fully understood? Observational study including 915 consecutive patients who underwent selective duct excision. Breast Cancer Res Treat 2010; 123:895-900
[7]: Stafford AP, De La Cruz LM, Willey SC. Workup and treatment of nipple discharge—a practical review. Annals of Breast Surgery 2021; 5
3) Page 1, line 42. Please expand on what is meant by potential for misdiagnosis and difficulties in imaging evaluation.
Reply: This line has been removed in a re-organized introduction (see point (2)) to reduce ambiguity.
4) Page 1, line 44. A patient with symptomatic BND will typically undergo biopsy or surgical consultation, for both benign pathology (such as papilloma) and malignancy, in efforts to find resolution for the patient’s BND. The introduction should create an argument for how recognition of benign pathology on CEM will aid in patient management. Even with benign pathology, patients with BND will undergo surgery to resolve symptoms.
Reply: This portion has been mentioned in the manuscript in lines 57-60: “It is hence imperative to differentiate benign and pathological causes of BND for early detection of breast cancers and to avoid unnecessary surgical interventions for patients with benign causes.”
In our institutions, surgical excision may not be warranted in the event of benignity. These cases can be followed up as bloody nipple discharge may resolve/clear-up on its own. However, the authors agree with the reviewers’ comment that in cases of persistent BND, patients would undergo surgery to relieve symptoms. However, in our experience, majority of Asian patients/women would prefer non-invasive follow-up in the event of benignity. This could be a result of difference in health-seeking behaviour among Asian women.
These findings also concur with NCCN guidelines for nipple discharge (figure inserted below). In benign cases, patients can be followed-up with clinical and radiological examinations if symptoms stabilize or resolve. Hence, no further changes have been made in the manuscript.
5) Page 2, lines 47-56. This paragraph better describes the argument for evaluating CEM as a modality for evaluation of BND. I would suggest eliminating the prior surgical argument and expanding on the limitations of FFDM, US, MRI, and galactography.
Reply: In the re-organized introduction, this paragraph starts off on its own without linking to the prior surgical argument. The limitations of FFDM, US, MRI and galactography have been expanded according to the reviewer’s comment. The changes to the paragraph are reflected in bold and strikethrough in lines 61-73:
“Diagnostic imaging plays an essential role in evaluation of BND and often include full-field digital mammography (FFDM), ultrasound (US), ductography (also known as galactography) and Magnetic resonance imaging (MRI). However, each of these modalities has its limitations. .Conventional diagnostic imaging plays an essential role in evaluation of BND ; however, it exhibits inherent limitations. Ultrasound has limited sensitivity (56%) and specificity (75%) in detecting intraductal lesions associated with BND may lead to the failure in identifying small or early-stage malignancies [8]. FFDM and US often show negative results in up to 50% of patients with bloody nipple discharge [9]. While contrast-enhanced MRI shows higher sensitivity as compared to FFDM and US, it is deemed not cost-effective and generates additional false positive findings [10]. Ductography is technically challenging and is a potentially uncomfortable procedure for patients with reported failure procedure rates of 15-23% [9]. There is hence an unmet need for an optimal imaging modality for the evaluation of BND.”
[8]: Bahl M, Baker JA, Greenup RA, Ghate SV. Diagnostic Value of Ultrasound in Female Patients With Nipple Discharge. AJR Am J Roentgenol 2015; 205:203-208]
[9]: Berger N, Luparia A, Di Leo G, et al. Diagnostic Performance of MRI Versus Galactography in Women With Pathologic Nipple Discharge: A Systematic Review and Meta-Analysis. AJR Am J Roentgenol 2017; 209:465-471
[10]: Baltzer PAT, Kapetas P, Marino MA, Clauser P. New diagnostic tools for breast cancer. Memo 2017; 10:175-180].
6) Page 2, line 57. Review and edit style and word choice. For example, emerging and new are similar descriptors, so eliminate one of those words. Similar examples throughout (example page 2, line 60, population and groups).
Reply: The authors agree with the comments. Words have been removed to reduce repetitiveness. In line 74, the statement now reads: “Contrast-enhanced mammography (CEM), an emerging technique..”. Similarly, in line 77, the statement now reads as: “There is preliminary evidence from non-asian population.”.
7) Page 2, line 62. BND, alone, is a reason for surgical intervention even with benign pathology.
Reply: This point has been addressed. See point (4) above.
8) Page 2, line 67. Please specifically define the goal of this study.
Reply: The authors have added more specifics to define the goal of the study. The statement in lines 84-86 now read as:
“Hence, the goal of the study is to evaluate the feasibility and role utility of CEM in evaluation of BND in the Asian population – in terms of its sensitivity, specificity and diagnostic accuracy.”
9) Page 2, line 71. Patients with BND should not undergo screening mammography, as these patients are symptomatic. Please explain why screening CEM exams were included in the search for the study population.
Reply: The authors would like to explain the difference in health screening protocols among different countries. Screening-detected breast cancers in Taiwan included both symptomatic and asymptomatic women, as biennial mammograms are offered free of charge to eligible women.
REF: Han, HJ., Chu, YC., Wang, J. et al. Characteristics of breast cancers detected by screening mammography in Taiwan: a single institute’s experience. BMC Women's Health 23, 330 (2023).
Hence, screening CEM exams were included in the search for the study population.
10) Page 2, Patient selection. Duplicate/repeated phrases, such as “for both institutions/both centers”. I would suggest decreasing word count and including high yield information more succinctly.
Reply: The authors thank the reviewer for the recommendation and have deleted the phrase “for both centers”. The statement in line 99 now read as:
“The cohorts were gathered systemically during the study period for both centers, and a total of 810 CEM studies were collected from these two institutions.”
11) Page 2, line 87. CEM was performed at two institutions, so briefly state if both institutions utilized the same vendor equipment and protocol.
Reply: The centers used same vendor equipment and protocol. A brief statement has been added in lines 118-119:
“The image acquisition of all 4 views was completed within 7 min. Both institutions utilized the same vendor equipment and protocol.”
12) Page 2, line 93. The protocol states CEM was performed first on the side of the “lesion of concern”. Was CEM performed first on the symptomatic side with BND or was there already a known “lesion of concern” prior to performing CEM?
Reply: The authors apologize for the confusion caused by the statement. It has now been amended in lines 113-114 to:
“Mediolateral oblique (MLO) and craniocaudal (CC) views of the symptomatic breast would be obtained first followed by CC and MLO views of the contralateral breast.”
13) Page 3, line 101. Was interpretation done by a radiologist or consultant? Please clarify credentials of the interpreter
Reply: The authors have amended, and at lines 122-125, the statement now reads as:
“Image analysis of the FFDM, US and recombined CEM images was done by two breast radiology consultants with at least 10 years of experience in the field of breast imaging.”
14) Page 3, CEM image interpretation and analysis paragraph. I would suggest reviewing these two paragraphs for style, word choice, grammar and punctuation. Similar recommendation for the “Pathology diagnosis” paragraph on page 3, line 117.
Reply: The authors have amended the paragraphs after consultation of professional medical writer from National University of Singapore (NUS).
The paragraph for Pathology diagnosis in lines 143-148 now read as:
“In cases with histological confirmation (i.e. biopsy or surgical excision reports), correlations between CEM imaging parameters and histological results were performed to categorize outcomes as benign or malignant. In cases without histological confirmation, the presumed benign nature of these imaging findings was validated by clinical evaluations and imaging follow-ups lasting over 24 months, to ensure accurate confirmation of their true benign status.
The following changes have been made for the paragraph “CEM imaging interpretation and analysis” from lines 122-130:
“Image analysis of FFDM, US and the recombined CEM images was done by two breast radiology imaging radiology consultants with at least 10 years of experience in the field of breast imaging. ; the final diagnosis was reached by their agreement (upon achieving consensus). The final diagnosis of the CEM images was reached upon consensus.
Image interpretation was updated performed according to the Contrast enhanced mammography (CEM) Breast Imaging-Reporting and Data System (BI-RADS) lexicon 2022. , considering lesion morphology, degree of enhancement, and distribution. The assessment began by detecting enhancing lesions and classifying them as either mass or non-mass enhancements with assessment of its conspicuity.”
15) Page 3. Table 1. The majority of patients with BND have benign pathology, as you also mentioned in the introduction. Please explain why your population of BND patients is 50% malignancy. This would suggest a bias in the study population.
Reply: The primary objective of the study is to assess the diagnostic efficacy of Contrast-Enhanced Mammography (CEM). Since CEM requires the injection of an intravenous contrast agent, which is a minimally invasive procedure, women with mild symptoms (e.g. single episode of slight bloody nipple discharge, spontaneous resolution of bloody discharge) may not be keen for CEM as a further investigation.
In addition, patients with history of papilloma which predispose them to bloody nipple discharge were also not eligible for the study (this is added in lines 102-103 under patient selection).
Overall, this has resulted in a selection bias towards women with more severe symptoms and also women with a higher pre-test probability. This explains the higher incidence of malignancy in our study (enriched cohort).
The following table shows two types of cohorts used in the evaluation of diagnostic tests: the Prevalence Cohort, which reflects the true prevalence of a condition in the general population, and the Enriched Cohort, which is optimized for the proportion of cases and controls to accurately assess diagnostic performance.
Feature |
Prevalence Cohort |
Enriched Cohort |
Purpose |
Reflects true prevalence of condition in the general population |
Optimizes the proportion of cases and controls for accurate assessment of diagnostic performance |
Necessity |
Not always necessary for comparing diagnostic performance |
Often used to ensure sufficient true positives and true negatives for reliable calculation of sensitivity and specificity |
Focus |
Determining prevalence of a condition |
Determining sensitivity and specificity of a test |
16) Page 4, line 145. Please clarify if the 4 cases who had follow-up only had persistent BND for the follow-up time period.
Reply: The authors would like to affirm these 4 cases had no persistent BND for the follow-up period. The following statement has been added in line 175: “All 4 patients had no persistent BND at their latest follow-up.”
17) Page 5, Figure 1. (A) states no suspicious finding on mammogram but then describes a focal asymmetry. Please make certain to ensure accuracy of figure descriptions.
Reply: The authors apologize for the misunderstanding. The following amendments have been made to lines 195-196:
“Bilateral mammogram (MLO view): No suspicious findings mass or microcalcifications seen.”
18) Page 7, line 180. For the 11 cases that were non-enhancing, please clarify what modality identified a lesion. Where any of these papillomas that were the cause of nipple discharge? Did they require excision to resolve the BND? As a reader, I would like to know how many of the benign cases of BND had resolution of symptoms during this study period.
Reply: Ultrasound or MRI were the modalities used to identify lesions in these 11 cases. Specifically, it helped detect dilated ducts and any suspicious intraductal lesions or masses associated with these ducts. Following this, ultrasound-guided biopsies and, where necessary, surgical excisions were performed to establish a definitive diagnosis.
Among the benign pathology cases (n=29), 17.2% of patients (5/29) underwent surgery, while 82.8% of patients (24/29) were managed conservatively with routine follow-up care. Of these 24 patients, 83.3% (20/24) showed resolution of their symptoms while the remaining 16.7% (4/24) had intermittent symptoms. We have included this information in lines 186-190.
19) Page 7, Table 2. Again, the large number of spiculated/irregular masses and segmental NME in this study population suggests possible selection bias, given that most cases of BND are typically benign. Did the study population have standard of care diagnostic FFDM and US prior to CEM? Is it possible that cases from the institutions were more likely to have malignancy based on imaging findings prior to CEM, and that is why CEM was performed? If so, please clarify in methods and patient selection. This would also explain the larger size (>1.5 cm) of lesions in this study. Lesions greater than 1.5 cm are typically seen on ultrasound, so it would help the reader if the methods of patient selection were clarified.
Reply: The authors have addressed the possible selection bias in point (15) above. The authors agree that larger lesions (>1.5cm) can be typically seen on ultrasound. However, in cases of non-mass lesions, up to 31% of lesions had no sonographic correlate and could only be detected on MRI [REF1].
A similar scenario may also occur with CEM [REF2].
[REF1]: Meissnitzer M, Dershaw DD, Lee CH, et al. Targeted ultrasound of the breast in women with abnormal MRI findings for whom biopsy has been recommended. AJR Am J of roentgenol. 2009;193:1025-9.
[REF2]: Huang PY, Tsai MY, Huang JS, Lin PY, Chou CP. Contrast-enhanced ultrasound-guided biopsy of suspicious breast lesions on contrast-enhanced mammography and contrast-enhanced MRI: a case series. J Med Ultrason (2001). 2023 Oct;50(4):521-529.).
20) Page 7, line 189. Low BPE level being a predictor of malignancy is contradictory to high BPE increasing likelihood of malignancy. Could the low BPE level be a correlation because lesions are easier to detect on CEM with low BPE. This would be an item for discussion, and possible limitation of CEM.
Reply: This is already discussed in discussion under the section on BPE from lines 436 to 448.
21) Page 9. I would limit results in this manuscript for additional findings, such as CEM size correlation with final pathological size and un-enhanced mammographic morphologic predictors of disease. The results section should focus most attention on the primary and dominant secondary endpoints.
Reply: The authors agree with the reviewer’s comment. The entire section “CEM Size vs Histological Size at T staging” has been removed from the manuscript. (See lines 285 to 295).
22) Page 10, line 268. When did the study patients have mammogram and ultrasound in relation to the timing of the CEM? Who interpreted the mammogram and ultrasound? The consultants or were charts reviewed? This portion of the study was not detailed in the Methodology section and should be described. Also, clarify whether screening or diagnostic mammography was performed (prior comment regarding screening versus diagnostic CEM as well).
Reply: As CEM is an add-on investigation on top of routine clinical investigations, the timing of CEM is hence performed after routine mammogram and ultrasounds.
The authors have amended this statement under Methodology in lines 92-93:
“All patients underwent diagnostic FFDM, and ultrasound followed by CEM examination — either CEM examinations were requested as part of clinical investigations or of a previously performed prospective study.”
The authors have amended the statement under “CEM image interpretation and analysis” in lines 122-123:
“Image analysis of FFDM, US and the recombined CEM images was done by two breast radiology imaging radiology consultants”
23) Page 10, line 282 and 284. I would suggest using BI-RADS terminology throughout. For example, mass rather than nodule.
Reply: The authors agree with the reviewer’s comment. The words have been replaced as shown (Lines 315 to 317):
“The last case was a benign appearing nodule mass on US with well-circumscribed margins deemed as a BIRADS 3 nodule lesion.”
However, CEM demonstrated surrounding non-mass enhancement around the nodule mass and the lesion was subsequently biopsied to represent DCIS.”
24) Page 11, line 303. There are a high number of malignant lesions in this study population, and this should be addressed in this study and any limitations to study design disclosed.
Reply: This has been addressed in points (15) and (19) above.
25) Page 11, line 305. One of the most important study results gleaned from this study population of symptomatic patients with BND is that all non-enhancing findings were benign. I would suggest expanding on this important detail. Surgical excision may still need to be performed due to symptoms, but it is helpful to know the high likelihood of benignity. Unfortunately, the selection bias of this study may hinder results as there are not a sufficient number of benign cases to make strong conclusions.
Reply: The authors appreciate your insightful feedback. This initial research focuses on uncommon breast conditions with specific symptoms, and we acknowledge the limitations of our current sample size. We are actively increasing the number of patients at two different medical centers to strengthen our findings and improve the assessment of how well we can diagnose BND. Future studies will help us pinpoint and understand potential limitations and pitfalls. We have also added your comment to the limitations section of our manuscript to address these concerns (see point (28) below).
26) Page 11, line 326. Surgical management is appropriate for BND, even in benign cases to resolve the BND. Conservative management with imaging only follow-up may not be appropriate if BND is not resolved. The conclusions drawn here may be inappropriate.
Reply: This has been addressed in point (4) above.
27) Page 12, lines 340-355. I would consider deleting this paragraph which does not discuss the implications of results from this study.
Reply: The authors have deleted the paragraph titled ““CEM vs Other Modalities (Ductography and MRI)” according to reviewer’s suggestion.
28) The Discussion should be significantly shortened and reviewed for relevance and discuss only conclusions that can be drawn from this study as well as for style, grammar, and repetitive topics. The discussion should provide the primary and secondary findings of this study, comparison of this study to other relevant literature, and discussion of limitation and future research. This discussion section draws several conclusions unrelated to the study results.
Reply: The discussion section has been significantly shortened and re-wrote according to the reviewer’s feedback. The discussion section is shown below (lines 333-482), with significant changes made in bold and strikethrough:
“Summary of important results
The study involved 58 patients; with 54 cases showing histological correlation. Out of the total, 50% of lesions were malignant and the sensitivity, specificity, PPV and NPV of CEM in evaluation of BND are 100%, 62.1%, 72.5% and 100% respectively. Ductal carcinoma in situ (DCIS) was found in 16 out of 58 cases (27.6%), and papilloma in 12 out of 58 cases (20.7%). The study categorized 11 out of 58 cases (19.0%) as non-enhancing and all cases were benign, while 47 cases (81.0%) showed enhancement, with 61.7% of these being malignant. The optimal cut-off value for lesion size on CEM for detecting breast cancer was determined to be 1.5cm (sensitivity 73.5 %, specificity 84.3%). 1.55cm (sensitivity 72.4%, specificity 86.2%, Youden index 0.586) and 1.75cm (sensitivity 69.0%, specificity 89.7%, Youden index 0.586). However, a smaller CEM size of 1.5 cm showed a higher sensitivity of 73.5 % with a corresponding specificity of 84.3%. These results suggest that clinically applicable size criteria with higher sensitivity and reasonable specificity for discrimi-nating between benign and malignant CEM contrast enhancement in BND patients could be 1.5 cm. Moreover, a 1.5 cm threshold is more easily retained and implemented in clinical practice. Significant predictors of malignancy on multivariate analysis include enhancing lesion of size ³ 1.5 cm (p-value 0.025) and suspicious morphological features (irregular/spiculated margins, irregular shape, segmental/linear NME distribution) (p-value <0.001). CEM BIRADS demonstrated excellent diagnostic performance with an AUC of 0.810.
Benefits of CEM and Clinical Applications
Many imaging modalities have been studied in the evaluation of BND but each has its own limitation [5-7]. There is hence an unmet clinical need for an optimal and cost-effective imaging modality for evaluation of BND to reduce the need for invasive surgical interventions. Our results demonstrated that CEDM CEM can be a very useful adjunct tool in assessment of BND. We found that patients which showed negative CEM results (i.e. no enhancement) had a negative predictive value of 100% in evaluation of BND. This could imply that these less suspicious patients can be reassured and managed conservatively with routine imag-ing/clinical follow-up with no further need of surgical interventions to achieve histological diagnosis. This will be helpful in cases of negative imaging findings or in cases with probably benign (BIRADS-3) findings. However, conventional diagnostic methods for BND are still necessary for BND patients with a high clinical suspicion of malignancy even if the findings on CEM are negative or benign. CEM also demonstrated higher sensitivity, specificity and diagnostic accuracy as compared to traditional modalities (combined MMG and US) (Table 4). In our study, CEM as an adjunct was able to detect 3 false negative cases from combined MMG and US (Two mammographic/sonographic occult cancers and one cancer upgrade from a BIRADS 3 lesion). The additional details on lesion morphology and vascular patterns on CEM can offer a more comprehensive assessment in addition to routine imaging modalities. These details can improve diagnostic accuracy and hence be helpful in risk stratification (benign vs malignant) of breast lesions in patients presenting with BND.
CEM’s high negative predictive value (NPV) of 100% could aid in risk stratification of patients presenting with BND. This could imply that these less suspicious patients can be reassured and managed conservatively with routine imaging/clinical follow-up, reducing the unnecessary further need of surgical interventions to achieve histological diagnosis. This will be helpful in cases of negative imaging findings or in cases with probably benign (BIRADS-3) findings. However, conventional diagnostic methods for BND are still necessary for BND patients with a high clinical suspicion of malignancy even if the findings on CEM are negative or benign.
In contrast, imaging features on CEM that raise suspicion for malignancy (e.g. high/moderate lesion conspicuity, irregular/spiculated margins of enhancing masses and segmental/linear distribution of non-mass enhancing lesions) would prompt further interventions to achieve a histological diagnosis. These imaging features would also boost the confidence of diagnostic radiologists when assessing for imaging-pathology concordance after biopsy results to reduce potential false negatives.
Ultimately, the authors would like to state that MRI remains the gold standard and excels in providing comprehensive 3D breast imaging and dynamic scanning sequences. However, it is often limited by its availability (long waiting time), long procedure time, and its high costs render this a non-cost-effective study to be performed for all patients with BND. On the other hand, CEM has a much shorter imaging acquisition time and can often be performed in the same patient visit to allow a one-stop work-up. Hence, CEM could represent a more logistical and cost-effective alternative for patients presenting with BND.
CEM vs Other Modalities (Ductography and MRI)
Although CEM demonstrates greater sensitivity and accuracy compared to conventional mammography and ultrasound in our study, ductal mammography currently remains the gold standard for evaluating BND due to its exceptional precision in delineating ductal anatomy and pathology [16] However, ductography is a challenging procedure with high-failure rates. It is also a time-consuming and potentially uncomfortable procedure for patients. Hence, alternatives such as contrast enhanced MRI has been recommended with reported sensitivities of 86-100% [17, 18] However, MRI is often limited by its availability (long waiting time), long procedure time, and its high costs render this a non-cost-effective study to be performed for all patients with BND.
Hence, CEM offers a more cost-effective and comfortable procedure on balance for patients presenting with BND. However, while CEM could be more cost-effective and simpler to implement than contrast-enhanced MRI, MRI excels in providing comprehensive 3D breast imaging and dynamic scanning sequences. These variations indicate the relevance of selecting the technique for diagnosis depending on individual clinical conditions.
Lesion conspicuity by radiologists
For lesions which enhance, CEM demonstrated a respectable PPV of 62.0%. Predictors for malignancy include lesion conspicuity, margins of masses and distribution of non-mass enhancements (NME). For lesion conspicuity, we found that lesions which demonstrate higher conspicuity had a higher chance of breast malignancy as compared to lesions which demonstrate low conspicuity (P-value = 0.019). This is likely based on the phenomenon that neoplasms induce angiogenesis for further tumor growth [22]. It is important to note that the current assessment of conspicuity in our study (based on ACR guidelines) remain subjective and further studies with use of quantitative measures or software could further help to improve malignancy risk stratification in the future [23].
Imaging features
Secondly, Worrisome features on CEM such as irregular margins (for enhancing masses) and worrisome distribution (linear/segmental for NMEs) show increased chance of breast malignancies in patients presenting with BND. This is largely unsurprising as these are identical predictors of malignancy on breast MRI. These features are readily apparent in sizeable lesions on CEM but can be difficult to fully characterize in lesions which are smaller (<1cm). Smaller lesions were also more likely to represent benign lesions in our study. We performed a Receiver Operator Characteristic (ROC) analysis and demonstrated that the optimal size cut-off in this study was 1.5cm. Lesions ≥1.5cm showed the high sensitivity and appropriate specificity for malignancy detection, with 21 out of 25 cases (84.0%) being identified as malignant.
BPE
To fully evaluate lesions on CEM, it is ideal to evaluate the lesion in its entirety with no obscuration. These lesions are typically best evaluated in cases with minimal-mild BPE as moderate-marked BPE has been well-known to hinder accurate evaluation on contrast enhanced modalities such as MRI [24]. Greater BPE has been reported to have a higher probability of developing breast cancer in high-risk women [25]. However, in our study, we did not find any relationships between moderate-marked BPE on CEM with breast malignancy. In contrast, detection of malignancy was significantly higher in cases with low (minimal or mild) BPE levels, as compared to their counterparts with higher (moderate or marked) BPE levels in our study. This is likely due to improved visibility of lesions on low BPE. The authors suggest that the relationship between BPE and malignancy may differ across specific clinical situations and patient groups. Further studies are warranted to understand the implications of BPE in patients presenting BND.
Clinical applications (combined with Benefits of CEM above to reduce repetition)
Our study demonstrates the efficacy of CEM as an initial screening tool for patients with breast nipple discharge due to its high negative predictive value that allows for conservative management and reduces the need for surgical interventions. Imaging features that raise suspicion for malignancy would include high/moderate lesion conspicuity, irregular shape of enhancing masses, irregular/spiculated margins of enhancing masses and segmental/linear distribution of non-mass enhancing lesions. By integrating the size threshold of 1.5 cm with these imaging characteristics, radiologists can enhance diagnostic accuracy.
Limitation
Our study has a few limitations. Firstly, our small sample size restricts the statistical power of our analysis and precludes the use of multivariate analysis. Additionally, the homogeneity of our sample may limit the diversity of perspectives represented, potentially overlooking important nuances or variations within the population. Secondly, there is likely a selection bias in our cohort with a significant proportion of malignant lesions identified (50%). As CEM is a minimally invasive procedure with injection of iodinated contrast, women with mild symptoms would be less willing to undergo a minimally invasive procedure for a trivial condition. Therefore, there is a natural selection bias towards willing patients with more severe symptoms who are more likely to present with breast malignancies. However, the enriched cohort study was focused on the utility of CEM in assessment of BND and an equal distribution of malignant and benign conditions may have been useful in improving the understanding of CEM appearances for both benign and malignant causes of BND. The authors acknowledge that the cohort in this study may not be representative and are currently actively increasing the number of patients at two different medical centers to strengthen our findings. Future studies will help us pinpoint and understand potential limitations and pitfalls. Thirdly, final histology was not readily available for all cases (as in cases with radiologically benign features) and we were unable to assess the ability of CEM to evaluate in situ lesion dimensions, since available pathological data were discontinuous. Future research endeavors should aim to replicate our findings with larger and more diverse samples to validate and extend the conclusions drawn from this study may also include comparison of MRI and CEM for BND when conventional diagnostic mammogram and ultrasound are negative. Despite these limitations, our study serves as a crucial foundation for further investigations into this area of inquiry.”
[22]: Heywang SH, Wolf A, Pruss E, Hilbertz T, Eiermann W, Permanetter W. MR imaging of the breast with Gd-DTPA: use and limitations. Radiology 1989; 171:95-103
[23]: Rudnicki W, Heinze S, Niemiec J, et al. Correlation between quantitative assessment of contrast enhancement in contrast-enhanced spectral mammography (CESM) and histopathology-preliminary results. Eur Radiol 2019; 29:6220-6226].
[24]: Hambly NM, Liberman L, Dershaw DD, Brennan S, Morris EA. Background parenchymal enhancement on baseline screening breast MRI: impact on biopsy rate and short-interval follow-up. AJR Am J Roentgenol 2011; 196:218-224
[25]: Dontchos BN, Rahbar H, Partridge SC, et al. Are Qualitative Assessments of Background Parenchymal Enhancement, Amount of Fibroglandular Tissue on MR Images, and Mammographic Density Associated with Breast Cancer Risk? Radiology 2015; 276:371-380
29) Page 13, line 406. Future endeavors may include comparison of MRI and CEM for BND when conventional diagnostic mammogram and ultrasound are negative.
Reply: This has been added into the manuscript in lines 477 to 480:
“Future research endeavors should aim to replicate our findings with larger and more diverse samples to validate and extend the conclusions drawn from this study. may include comparison of MRI and CEM for BND when conventional diagnostic mammogram and ultrasound are negative. Despite these limitations, our study serves as a crucial foundation for further investigations into this area of inquiry.”
30) Page 13, line 411. The conclusion made regarding clinical utility of CEM for BND may not be appropriate because both benign and malignant BND may be managed surgically.
Reply: This has been addressed in point (4) above.
We hope that the revisions that we have done and our response to the comments are satisfactory. Thank you very much and hope to hear from you soon.
